# Effects of Ambient Humidity on Water Migration and Hydrate Change in Early-Age Hardened Cement Paste

**DOI:** 10.3390/ma15248803

**Published:** 2022-12-09

**Authors:** Dafu Li, Bo Tian, Kaimin Niu, Lihui Li, Lei Quan, Xuwei Zhu

**Affiliations:** 1Research Institute of Highway Ministry of Transport, 8 Xitucheng Road, Haidian District, Beijing 100088, China; 2School of Civil Engineering, Chongqing Jiaotong University, Chongqing 400074, China; 3Key Laboratory of Road and Traffic Engineering of the Ministry of Education, Tongji University, Shanghai 201804, China

**Keywords:** relative humidity, NMR, XRD, FTIR, water migration, hydrogrossular

## Abstract

Ultra-low humidity environments will lead to changes in the microstructure of C–S–H, which will reduce the mechanical properties and service life of cement-based concrete. Thus, to further explore the mechanism on the microscale, this paper studied the water migration and the changes in the hydration products in white cement that was cured for 7 days at 20 °C and at different ambient relative humidities (RHs). The migration and transformation of different types of water in cement paste were studied by low-field nuclear magnetic resonance (NMR). At the same time, Fourier transform infrared spectroscopy (FTIR) and X-ray diffraction (XRD) were used to analyze semi-quantitatively the crystal phase in the hydration products. The results showed that in the first 7 days of the curing process, the content of the different types of water and the hydration products in the cement samples were influenced by the ambient RH. The total water content of the samples will decrease with the decrease in the RH; when the RH decreases to 54% or below, the chemically bound water in the samples will increase with the decline in the RH. Additionally, when the ambient RH is lower than 54%, the grossular will gradually transform into hydrogrossular crystals with the decrease in the RH, and the hibschite with less chemically bound water will transform into katoite with more chemically bound water. In future research, the water migration and hydrate changes under different curing ages, drying processes, and coupling effects should be explored.

## 1. Introduction

Cement-based concrete is one of the most commonly used building materials [1,2]. Due to its durability and excellent compressive properties, cement-based concrete has been used in many extreme environments [3,4]. The environment of the Qinghai–Tibet Plateau is one of the most extreme, with large temperature differences between day and night and low relative humidity (RH). Cement-based concrete structures in service on the Qinghai–Tibet Plateau need to face ultra-low humidity for a long time. Additionally, such a low RH environment will lead to the formation of a humidity gradient inside the concrete, which has a great impact on the migration of water in cement-based concrete [5]. Meanwhile, in cement paste, moisture exists in the C–S–H gel structures in various forms, and the moisture in the cement matrix was initially divided into mobile water and chemically bound water [6]. However, as the study on the C–S–H gel progressed, the mobile water in the cement matrix was further divided into capillary water, gel water, and interlayer water according to how closely they are bound with the C–S–H gel and where they exist. Pore water is mainly composed of neutral water molecules existing in the wider pores, and its binding with the cement matrix is not so close [7]. Gel water mainly exists in the C–S–H gel, which is a kind of specially adsorbed water. Interlayer water mainly exists between two C–S–H layers, and it will combine with cations to form hydrated ions. Since there are extra negative charges on the C–S–H layer, the cations will be adsorbed on the surface of the C–S–H layer structure and continue to adsorb water molecules until they finally form the interlayer water. Interlayer water has a great influence on the layered structure of C–S–H [8]. 

In a low RH environment, the different forms of water will convert to each other, and some of them will migrate to the external environment [9,10]. Such interconversion and migration will have a great impact on the mechanical properties of concrete structures [11,12]. Previously, F. Wittmann studied the volume shrinkage and elastic modulus of cement paste cured under different RH conditions [13]. In addition, B. Zech studied the elastic modulus and pore distribution of cement-based concrete under the coupling effects of temperature and RH conditions [14]. Their research determined that the compressive and flexural strengths of cement paste cured at lower ambient RHs showed a greater decline than that of cement paste treated under conventional conditions, and this decline would reduce the service life of cement-based concrete structures. To further determine the migration and interconversion of moisture in the cement paste under the condition of low humidity curing, Muller and Maruyama studied the migration process of water in C_3_S samples by using the low-field nuclear magnetic resonance (NMR) method [15,16]. They found that ambient RH would affect the chemical composition of C–S–H and determine the variations of the water forms and surface area of C–S–H. Moreover, neutron scattering and X-ray diffraction (XRD) experiments were also carried out [17,18,19]. The experimental results showed that water migration could lead to changes in the C–S–H pore structure, which will cause the water in the cement matrix to easily migrate to the external environment. The studies stated above include the following: the mechanical properties of low-RH hardened cement paste, the water migration process, and the changes in the C–S–H microstructure. However, studies on the changes in the hydration products of cement paste under ultra-low humidity conditions are still not thorough.

Based on the above research results, the water migration and the chemical composition changes of 7-day-old white aluminate cement were studied by low-field nuclear magnetic resonance (NMR), Fourier transform infrared spectroscopy (FTIR), and X-ray diffraction (XRD). Based on the Butler–Reed–Dawson (BRD) model and the full width at half maximum (FWHM) algorithm [20], the NMR and infrared spectral results were analyzed. The migration of non-chemically bound water and the interconversion of all of the water components (gel water, interlayer water, and chemically bound water) in cement paste under different RH conditions are discussed. Some hydration products were determined by FTIR and XRD, and the relationship between the decrease in interlayer water and the increase in chemically bound water was confirmed. The crystalline phase transition of the hydration products in cement paste under different RH conditions was studied.

## 2. Materials and Methods

### 2.1. Materials

Albo white Portland cement was used to prepare the cement paste; this cement was chosen to avoid the influence of iron in the OPC on the NMR and FTIR results. Its chemical composition is shown in Table 1. Additionally, the deionized water was bought from Wenzhou Hengtong Corporation, Zhejiang, China. 

### 2.2. Experiments

#### 2.2.1. Cement Paste Preparation

The white cement and deionized water stated above were used to prepare the cement paste with a water–cement ratio of 0.4. In order to ensure that the cement paste had better uniformity, a mixer was used to stir, and the stirring time was 5 min with a stirring speed of 300 r/min. After the cement was mixed evenly, it was injected into a silica gel mold with a size of φ5 × 2 cm. The sample was sealed immediately and kept under different RH conditions for 7 days for later use.

#### 2.2.2. Drying Process

The RH in the different drying dishes was controlled by a saturated salt solution, water-absorbing silica gel, or pure water, as well as it was monitored by a hemispheric hygrometer. Additionally, the drying dishes were placed in a constant temperature chamber for 1 day, ahead of the curing of the cement paste to ensure that the RH in the drying dishes can reach the designed value. After this, the samples were placed in the drying dishes, which were sealed with wax and plastic films (Figure 1a,c). The temperature was set at 20 ± 1 °C, which was controlled by a temperature and humidity controller, as shown in Figure 1b. 

Eight different RH conditions were designed in this paper (RH 5%, 23%, 43%, 54%, 75%, 81%, 85%, and 98%) [21]. The RH controllers and their corresponding ambient RHs are shown in Table 2. After curing for 7 days, the samples were demolded and ground with a ball grinder at a speed of 1000 r/min for 2 min and tested by XRD, FTIR, and NMR, respectively (Figure 2).

#### 2.2.3. X-ray Diffraction 

Rigaku ULTIMA IV was used for the XRD analysis. The 2θ scanning range was between 5 and 75 degrees, the scanning speed was 2 degrees per minute, and the scanning increment was 0.02 degrees. Besides, to analyze and calculate the content of each crystalline phase, the K-value method was used. The mass ratio of the crystalline phase was calculated according to Equation (1) as follows: (1)$$\frac{I_{x}}{I_{s}}=K\cdot \frac{w_{x}}{w_{s}}$$
where *x* is the crystal phase to be measured, *I_x_* is the main peak intensity of *x*, *s* is α-Al_2_O_3_, *I_s_* is the main peak intensity of *s*, and *w_x_* and *w_s_* represent the mass ratios of *x* and *s* in the samples, respectively. Additionally, the definition of *K* is that a certain pure substance, *x*, and α-Al_2_O_3_ are mixed evenly according to a weight ratio of 1:1, and the diffraction intensity ratio of the two is the *K* value. Here, “diffraction intensity” is defined as the height of the peak instead of the area [22]. 

#### 2.2.4. Fourier Transform Infrared Spectroscopy

The Nicolet iS50 spectrometer (ThermoFisher Scientific Inc.) was used to test the samples and obtain the FTIR results. The attenuated total reflection (ATR) mode was used to analyze the samples directly over a thickness of several microns and a sampling area of approximately 1 mm square. A diamond was used as the reflective crystal. The scans were performed in the range of 4000~400 cm^−1^, with a spectral resolution of 4 cm^−1^. Additionally, the results of the FTIR were fitted by the FWHM method to compare the contents of different hydration products [20]. 

#### 2.2.5. Nuclear Magnetic Resonance

The relaxation times of the samples were measured by a MesoMR23-060H-I medium-size NMR analyzer. The resonance frequency was 23 MHz, the magnet strength was 0.5 T, the echo time was 0.2 ms, and the π/2 pulse length was 23.04 μs. 

In this paper, the NMR results of the samples were measured by the Carr–Purcell–Meiboom–Gill (CPMG) pulse to obtain the T_2_ relaxation spectrum. The CPMG method was proposed by H.Y. Carr and improved by S. Meiboom and D. Gill to obtain a more accurate T_2_ relaxation spectrum [23]. The measured CPMG data were analyzed with the inverse Laplace transform (ILT) algorithm, and the Butler–Reed–Dawson (BRD) method was used to obtain a continuous T_2_ distribution. The ratio of each peak to the total intensity was calculated to determine the contents of the interlayer water, gel water, pore water, and chemically bound water [24]. 

The chemically bound water was quantified according to the method described in a previous study by Muller [15]. The water distribution in the samples and the ratio of free water to chemically bound water could be obtained by splitting the quadrature echo decay signals into gaussian and exponential signals, which represent solid and liquid echos, respectively. Thus, based on such a method, the mass ratios of all the water components in the paste samples were identified. The mass difference and the total solid echo signals were calibrated to obtain the water distribution per gram of cement.

## 3. Results & Discussion

### 3.1. NMR

According to the relaxation spectrum results, shown in Figure 3, the contents of the different forms of water (including pore water, interlayer water, gel water, and chemically bound water) in the sample were calculated, and four different peaks were obtained by using the BRD algorithm. The relative content of the different types of water after curing for 7 days is shown in Figure 4.

After calculating the T_2_ relaxation spectrum results of the samples, it can be concluded that when the curing time is 7 days, except for the sample cured at an ambient RH of 98%, the pore water content in the samples will decrease to below 0.01 g/g cement; thus, the change in pore water content will not be discussed in detail in this paper. 

As it can be seen from Figure 4, the total water content in the sample gradually decreased with the decline in the ambient RH. First, when the RH was reduced to less than 81%, with the continuous decrease in RH, the interlayer water content increased, with the highest value appearing at the ambient RH of 54%, and then the interlayer water content continued to decrease. Second, the content of chemically bound water barely changed when the ambient RH was higher than 54% and remained constant at about 0.1 g/g cement. However, when the samples were cured at an RH of 54% or below, the content of chemically bound water in the samples gradually increased with the decrease in the ambient RH. Under the conditions of RH 54%, 43%, 23%, and 5%, the increases in chemically bound water in the four samples were about 19%, 33%, 47%, and 66%, respectively, compared with those of the samples cured under an RH of 98% (0.09 g/g cement). This increase in chemically bound water was less than the decrease in interlayer water observed above. 

The conversion of the three types of water discussed above indicates that when the ambient RH is in the range between 54% and 98%, the RH of the pores is low, resulting in the migration of interlayer water to the external environment. At the same time, the content of gel water is still sufficient to convert to interlayer water while maintaining the microstructure and chemical composition of the C–S–H gel. Moreover, the conversion from gel water to interlayer water is easier than that of interlayer water to pore water, so the content of interlayer water will first increase with the decrease in the ambient RH. However, when the ambient RH is lower than 54%, the content of gel water is too low to provide more water to the interlayer structure of the C–S–H, nor can it continue to maintain the original gel form of the C–S–H [25]. At this time, in addition to migrating into the pores, the interlayer water will also be absorbed by the C–S–H during the migration due to the close approach of the C–S–H layer structure during the migration process. However, since the gel form of C–S–H is no longer stable, the absorbed water will be attached to unhydrated or incompletely hydrated cement particles to form new chemically bound water. Additionally, more of the interlayer water will escape to the external environment, so the increase in chemically bound water will be smaller than the decrease in interlayer water [26].

The results mentioned above indicate that with the curing RH decreasing, the gel water will first convert into interlayer water and then into chemically bound water after the ambient RH decreases to 54% or lower. The increase in chemically bound water suggests that the change in the ambient RH will affect the hydration process of cement-based materials to a certain extent. In this paper, the changes in the hydration products in the samples will be further discussed based on the results obtained by the XRD and FTIR. 

### 3.2. XRD

Figure 5 shows the XRD results of the samples cured at 20 °C at different ambient RHs for 7 days.

Table 3 shows the different *K* values of the hydration products. 

In addition, in Table 4, the results of the mass ratios of the hydration products are calculated according to the K-value method.

It can be seen from Table 4 that, except for the minerals of the hydrogrossular series, the mass fractions of the other hydration products selected in this paper, including ettringite (Ca_6_Al_2_(SO_4_)_3_(OH)_12_), portlandite (Ca(OH)_2_), akermanite (Ca_2_MgSi_2_O_7_), tricalcium aluminate (Ca_3_Al_2_O_6_), C_3_S (Ca_3_SiO_5_), and C_2_S (Ca_2_SiO_4_), do not change significantly with the change in RH in the external environment. The content of C_3_S is the highest, accounting for 35.3% of the hydration products. The mass fractions of portlandite and C_2_S are close, and their mass ratios are about 14.5% and 18.6%. In addition to the minerals of the hydrogrossular series, the content of ettringite, akermanite, and tricalcium aluminate is close, and their mass ratios are pretty low, which are about 6.7%, 9.7%, and 8.4%, respectively. It can be seen that the contents of the hydration products mentioned above barely change.

Hydrogrossular (Ca_3_Al_2_(SiO_4_)_3−x_(OH)_4x_) is one of the most important hydration products of cement-based materials. These minerals can be divided into many different crystals according to their number of hydroxyl groups, including hibschite, palzolite, grossularoid, katoite, grossular, etc. [27]. According to the XRD test results, strong intensity peaks of two crystalline phases, hibschite (Ca_3_Al_2_(SiO_4_)_3−x_(OH)_4x_, 0.2 < x < 1.5) and katoite (Ca_3_Al_2_(SiO_4_)_3-x_(OH)_4x_, x > 1.5), can be seen. The intensity peaks of the other crystalline phases are not prominent and overlap with each other; thus, the changes in these two crystalline phases will be mainly discussed in the following paper.

After processing the XRD spectrum results of the eight groups of samples, it can be seen that the main peaks of the minerals of the hydrogrossular series are concentrated in areas A and B, in which the main peaks of katoite (peak a), hibschite (peak b), and grossular (peak c) can be found, respectively. An independent intensity peak of Ca(OH)_2_ (peak d) can also be seen in region A. According to the calculation results in Table 4, the content of Ca(OH)_2_, C_2_S, and C_3_S in the samples does not change when the ambient RH changes, so the intensity of peaks a and b can be regarded as a reference for the relative content of katoite and hibschite. As shown in Figure 4, when the ambient RH decreases to less than 75%, the total content of hydrogrossular series products in the samples gradually increases with the decline in the ambient RH. However, the change in the intensity peak of the grossular (peak c) is opposite to that of the hydrogrossular. With the decrease in the ambient RH, the main peak of the grossular is still strong when the RH is higher than 75%. However, when the RH decreased to 75% or below, the intensity of its main peak gradually decreased until it almost disappeared. The reason for this phenomenon can be that there is still some grossular that is not involved in the hydration reaction in the internal C–S–H layer structure, but when the ambient RH decreases and causes the interlayer water migration in the samples, this part of the grossular will react with the water molecules adsorbed on the C–S–H surface. Then, new hydrogrossular (katoite and hibschite) will be formed.

At the same time, it can also be observed that the main peaks of katoite and hibschite shift with the change in the ambient RH in area A, as shown in Figure 5. When the RH declines to 54% or below, a gradual increase in the intensity of peak a can be seen, while the opposite is observed for peak b, which, to some extent, indicates the possibility of reciprocal conversion between the two crystals. The conditions for mutual transformation between these two crystals are not harsh, and reversible transformation can occur when the temperature of the external environment is 20 °C. Katoite will be dehydrated and transformed into hibschite with less chemically bound water when the temperature is above 20 °C [28]. It can be observed that when the external RH changes, the water in the samples will migrate and the mass ratio of the two crystals will change. In the experiment, the ambient temperature was set to 20 ± 2 °C, which is the critical temperature for the conversion of the two kinds of crystals, so that the crystal transition can be affected by the change in the ambient RH. When the ambient RH is above 75%, the free water provided by the external environment is still relatively sufficient. Under the same hydration time (7d), the content of chemically bound water will not change greatly, so the mass ratios of katoite and hibschite are relatively stable. However, when the RH decreases to 54% or below, the content of hibschite gradually decreases, while the content of katoite gradually increases, which is also consistent with the FTIR results. As explained above, grossular will gradually convert to hydrogrossular when the RH is below 54%, and this process is dominated by the increase in metal-bonded hydroxyl groups; the transformation process from hibschite to katoite is the same. Therefore, grossular will first convert into hibschite, which has fewer metal-bonded hydroxyl groups, and then hibschite will continue to transform into katoite, which also explains the shift and the change in the intensity of peaks a and b. According to the lattice parameter of the hydrogrossular crystals, as shown in Table 5 [29], when the RH decreases to 54% or below, the volume of C–S–H in the hardened cement paste will increase with the decrease in the humidity, which is consistent with the change in the C–S–H surface area calculated by Maruyama [19]. Therefore, the conversion of hibschite to katoite caused by the decrease in the environmental RH is one of the reasons for the increase in the C–S–H surface area.

The above results are consistent with the NMR measurements. The addition of chemically bound water and the results of the XRD proved that the hydration product will convert due to the change in the ambient RH, which further explains the reason why ambient RH will have an influence on the mechanical properties [30,31]. In addition to the crystal-type transition observed in this paper, there can be changes in other hydration products contributing to the increase in chemically bound water that are not observed by the XRD and FTIR results in this study. 

### 3.3. FTIR

Based on the crystal phase results determined by the XRD, the FTIR test was carried out on cement samples that were cured for 7 days under different ambient RHs. The results are shown in Figure 6.

Chemically bound water in cement hydration products mainly contains two parts: the crystal water of the compound and the metal- or silicon-bonded hydroxyl groups [32]. Through the spectral line results in the mid-infrared region (MIR) (Figure 6a,b), the diffuse bands or sharp peaks of most of the unbound water and the O–H bond strength peak of the chemically bound water can be directly observed to further determine the changes in chemically bound water, including crystal and hydroxyl water, in the sample. According to the corresponding frequencies of various chemical bonds, which are presented in Table 6, the FWHM method was used to fit the spectral line results, and the corresponding peak height of each frequency peak was calculated to determine the relative content of different chemically bound waters. 

In Figure 6, a diffuse band can be observed at about 3416 cm^−1^ overlapped with other peaks, which is caused by the tensile vibrations of the physically adsorbed water molecules on the surface, while at about 1640 cm^−1^, the band caused by stretching vibrations only shows a shoulder [33,34]. Thus, the content of the absorbed water cannot be compared directly with the intensity of the diffuse bands. 

As for the other chemical bonds, a strong and independent characteristic peak of Si–OH produced by the combination of -OH and [SiO_4_]^4−^ (≈1260 cm^−1^) can be seen in all of the sample spectral line results [35]. In the high-frequency region, the following flexural vibration intensity peaks are mainly calibrated: the Ca–OH bond in Ca(OH)_2_ at about 3642 cm^−1^ and in C_2_SH at about 3540 cm^−1^ [36,37]. These kinds of Ca–OH bonds will form an overlapped band between 3200 and 3670 cm^−1^. In addition, the γAl–OH bond in the hydrogrossular has a strong influence on the diffuse bands from 3620 to 3670 cm^−1^ and 620 cm^−1^ and forms a sharp intensity peak (≈3660 cm^−1^) near the overlapped band of Ca–OH in different crystals. When the RH is lower than 54%, this intensity peak will shift to a higher frequency with the decrease in the ambient RH. This shift can be considered as [SiO_4_]^4−^ in the grossular being replaced by −OH to form the minerals of the hydrogrossular series, including hibschite and katoite [38].

In order to further verify the observational results in the low frequency region, the FWHM method was used to fit each intensity peak, and the boundary of the diffuse band at 2000 cm^−1^ was taken as the divider. Figure 7 shows the fitting results for the RH 5% and 23% samples. Additionally, the fitting results of each intensity peak of the spectral lines of the other samples are shown in Table 7. 

In the results of the FTIR experiments, hibschite and katoite are further calibrated and distinguished in order to analyze the relative content changes of the two kinds of hydrogrossular crystals. The relative content of [SiO_4_]^4−^ in the samples is relatively stable because the Ca/Si ratio of the cement is exactly the same and the water–cement ratio is fixed [39]. 

Therefore, in this paper, the relative content of the γAl–OH bond of the minerals in the hydrogrossular series was calculated. The ratio of the peak height of γAl–OH to the peak height of Si–OH in [SiO_4_]^4−^ at 1260 cm^−1^ was used as an index to measure the relative content of the γAl–OH bond in the samples. The results are shown in Table 7. It can be seen that the content of the γAl–OH bond will not change significantly when the ambient RH is higher than 75%, but when the RH is lower than 54%, the intensity peak will increase with the decrease in the RH, which is also consistent with the observed changes in the content of hibschite and katoite in the XRD results. In conclusion, when the ambient RH is lower than 54%, [SiO_4_]^4−^ in the hydrogrossular will be gradually replaced by -OH, and more hibschite will convert to katoite due to the decrease in the ambient RH. This conclusion is consistent with the XRD results and also explains why the peak at 3660 cm^−1^ will shift to a higher frequency when the RH decreases. The above conclusions further show that when the ambient RH drops below 54%, the content of chemically bound water, especially the bonded hydroxyl groups, in the sample will be changed by the ambient RH, and the hydration products will also change with the decrease in the ambient RH.

## 4. Conclusions

This paper studied the morphology changes, the migration of water, and the changes in the hydration products in white cement samples cured at 20 °C and at different ambient RHs for 7 days. The cement samples were tested by NMR, XRD, and FTIR methods. According to the test results and discussion, the following conclusions were obtained: The three different forms of water show different trends when the ambient RH decreases. The content of gel water decreases continuously until there is almost no gel water left. In addition, the content of interlayer water first rises and then decreases, and the maximum value occurs when the ambient RH equals 54%. As for the content of chemically bound water, it hardly changes when the ambient RH is higher than 54%. However, when the ambient RH decreases to 54% or lower, the chemically bound water shows a different trend from the gel water and interlayer water trends; its content increases as the ambient RH decreases. This may be because gel water will first convert into interlayer water, then capillary water, and, finally, it will migrate to the external environment when the ambient RH decreases, and gel water plays an important role in the maintenance of the microstructure of C–S–H. Thus, as the ambient RH declines, when gel water is still sufficient to keep the microstructure stable, the content of chemically bound water will remain at around 0.09 g/g cement. However, when the gel water is too little and, at the same time, the interlayer water is moving to the wider pores, during which some of the water will be absorbed when attached to the surface of C–S–H, the unhydrated particles will react with the newly absorbed water to form new chemically bound water.When the ambient RH is higher than 75%, the content of hydrogrossular in the hydration products remains stable. However, when the ambient RH is lower than 54%, the content of hydrogrossular in the hydration products increases with the decline in the ambient RH. The contents of the two kinds of hydrogrossular, which are hibschite and katoite, will be influenced by the ambient RH. When the ambient RH is lower than 54%, the content of hibschite will gradually decrease while the content of katoite will rise. A possible reason is the existence of some unhydrated grossular in the C–S–H layer structure during the hydration process. When the migration of the interlayer water happens, some will be absorbed by the C–S–H, so the unhydrated grossular reacts with the absorbed water and forms hydrogrossular. As the interlayer water increases, the hibschite with less chemically bound water will gradually convert to katoite with more chemically bound water.

In future research, experiments on cement paste with a shorter curing time can be carried out to further study the specific process of water migration in cement under low humidity curing conditions. Additionally, the changes in the hydration products, except for hydrogrossular, which are affected by the curing condition of low humidity, should be studied. Last but not least, the migration process of water and the change in the hydration products in cement should be studied under the coupling action of various environmental factors, such as carbonization, large temperature differences, low humidity, etc.

## Figures and Tables

**Figure 1 materials-15-08803-f001:**
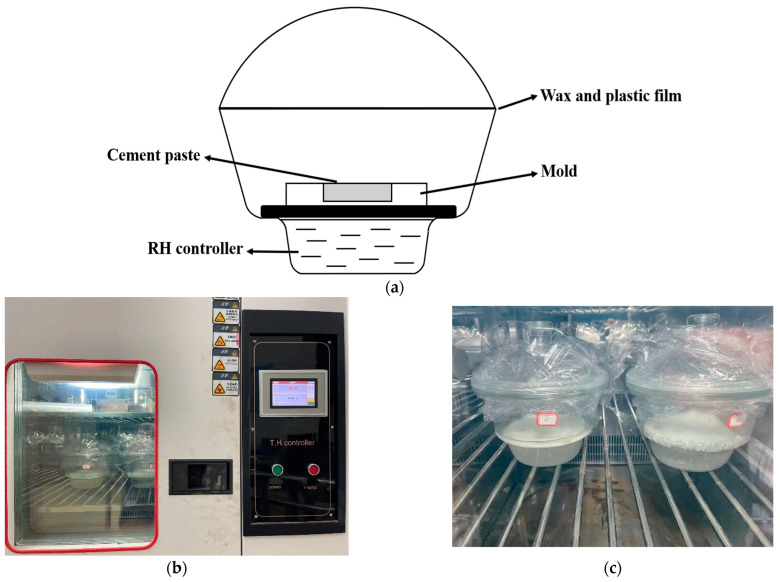
Curing of the samples under different RH conditions. (**a**) Schematic diagram of the drying dishes. (**b**) Temperature and humidity controller. (**c**) The drying dishes containing the cement samples and a saturated salt solution.

**Figure 2 materials-15-08803-f002:**
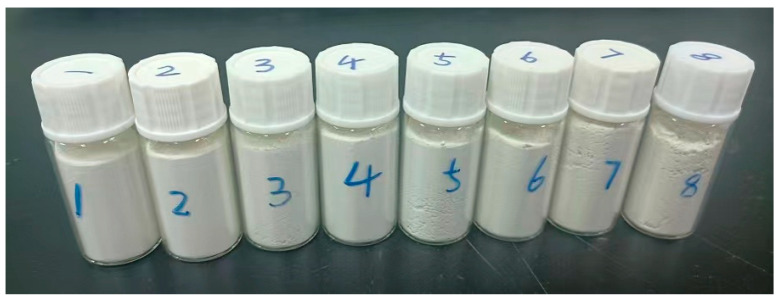
Samples prepared for the NMR, XRD, and FTIR tests (the samples cured at RH 5% to 98% were numbered in the sequence from 1 to 8).

**Figure 3 materials-15-08803-f003:**
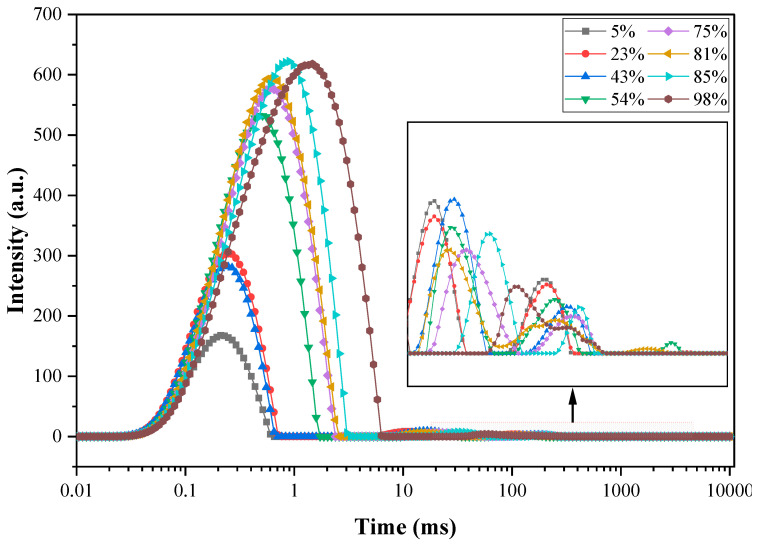
T_2_ relaxation spectra of samples cured at different ambient RH.

**Figure 4 materials-15-08803-f004:**
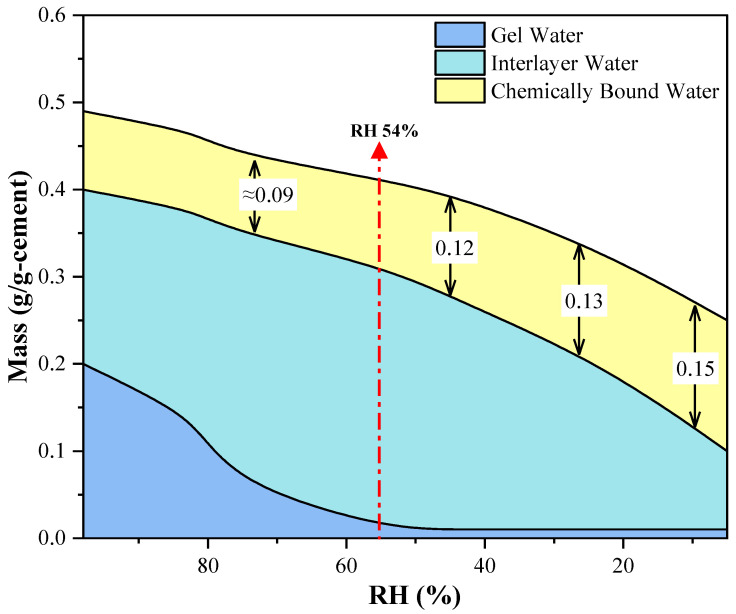
The calculated result of three types of water in cement paste.

**Figure 5 materials-15-08803-f005:**
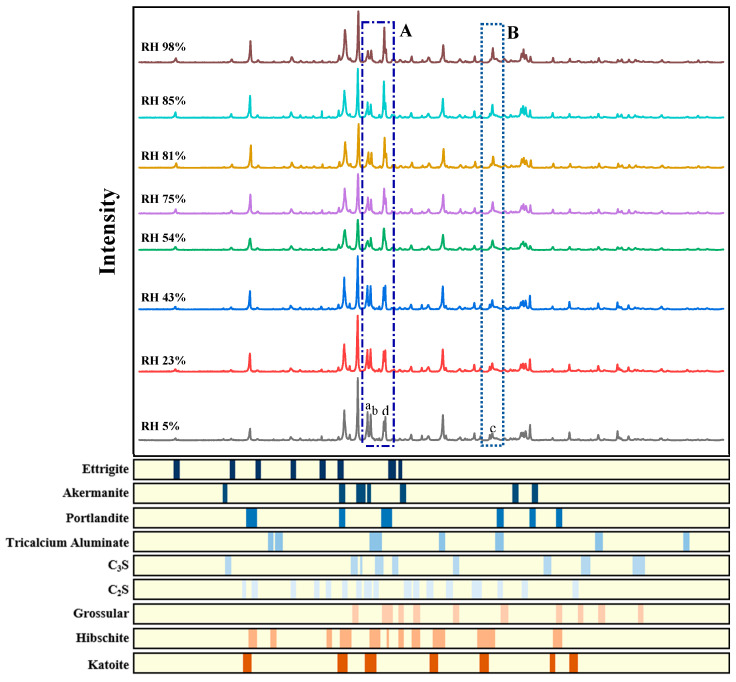
XRD patterns of the samples cured at 20 °C for 7 days at different ambient RHs. (peaks a, b, c, and d represent the peaks of katoite, hibschite, grossular, and portlandite, respectively, and the yellow bands and colored squares below indicate the location of the intensity peaks of each phase according to the ICDD-PDF database).

**Figure 6 materials-15-08803-f006:**
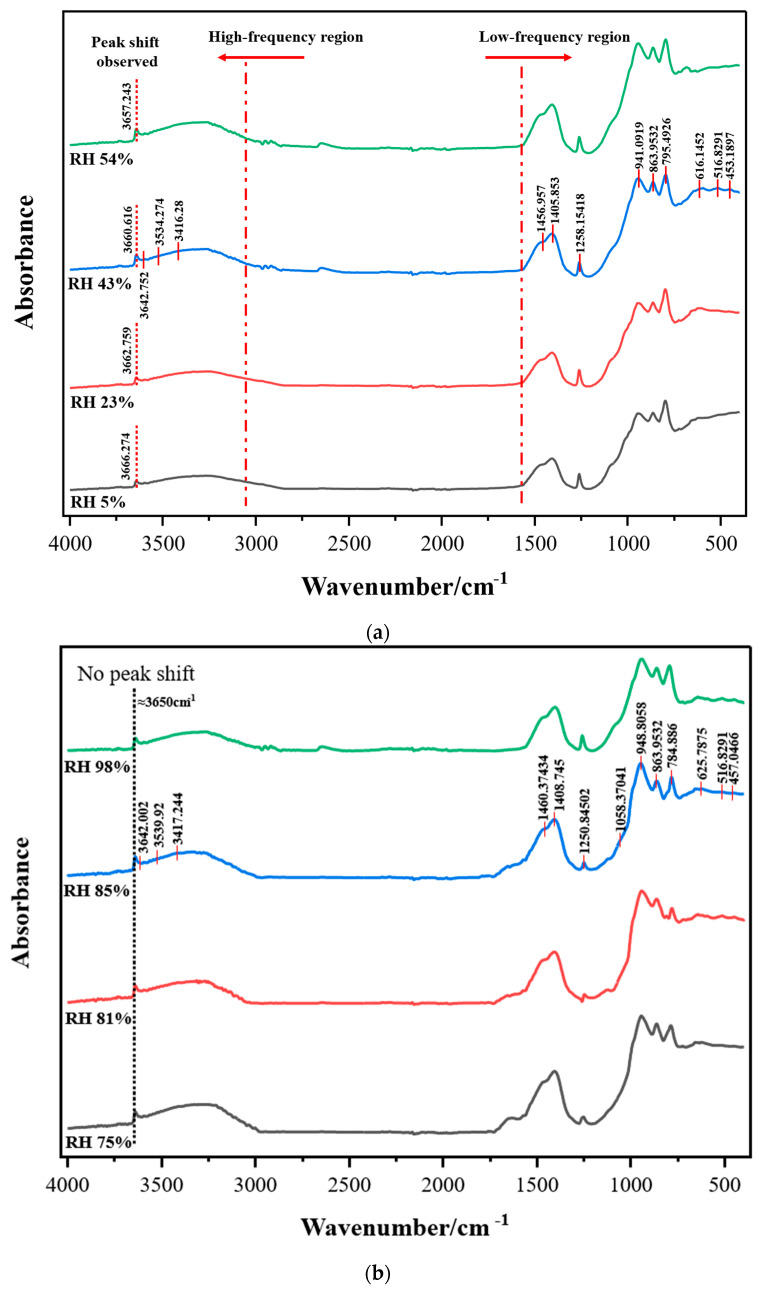
FTIR results of the samples cured at different ambient RHs. (**a**) Results of the samples cured at RHs from 5% to 54%; (**b**) results of the samples cured at RHs from 75% to 98%.

**Figure 7 materials-15-08803-f007:**
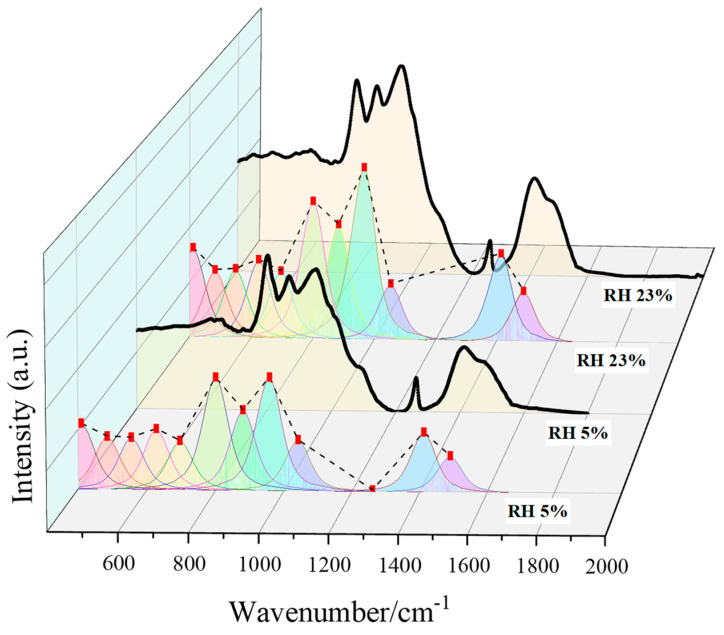
The fitting results of the FWHM of the samples cured at RH 5% and 23%.

**Table 1 materials-15-08803-t001:** The chemical composition of Albo white cement.

Chemical Composition	SiO_2_	Al_2_O_3_	CaO	MgO	Na_2_O	K_2_O	Fe_2_O_3_	SO_3_	LOI
Mass Ratio (%)	21.13	3.83	65.44	1.19	0.41	1.16	0.21	3.34	3.29

**Table 2 materials-15-08803-t002:** Saturated salt solution and water-absorbing silica gel for controlling the RH.

RH Controller	Water-Absorbing Silica Gel	CH_3_COOK	(CH_3_COO)_2_Ca	Mg(NO_3_)_2_	NaCl	(NH_4_)_2_SO_4_	KCl	Water
RH	5%	23%	43%	54%	75%	81%	85%	98%

**Table 3 materials-15-08803-t003:** K values of the hydration products at different ambient RHs.

Name	Ettringite	Akermanite	Portlandite	Tricalcium Aluminate	C_3_S	C_2_S	Hydrogrossular
K Value	1.75	2.53	3.69	3.27	0.8	0.76	2.35

**Table 4 materials-15-08803-t004:** Mass ratios of the hydration products of the cured samples.

RH	Mass Ratio (%)						
	Ettringite	Akermanite	Portlandite	Tricalcium Aluminate	C_3_S	C_2_S	Hydrogrossular
5%	6.94	9.78	14.44	8.42	35.03	18.44	6.95
23%	6.96	9.75	14.38	8.38	35.32	18.57	6.63
43%	6.63	9.81	14.78	8.41	35.27	18.67	6.42
54%	6.67	9.71	14.79	8.41	35.38	18.77	6.27
75%	7.03	9.83	14.72	8.39	35.32	18.61	6.11
81%	7.06	9.84	14.49	8.4	35.44	18.72	6.05
85%	7.03	9.92	14.57	8.45	35.20	18.79	6.03
98%	7.05	9.74	14.75	8.37	35.32	18.72	6.06

**Table 5 materials-15-08803-t005:** The lattice parameters of hibschite and katoite.

Name	Grossular	Hydrogrossular		
		Hibschite	Katoite	
Chemical formula	Ca_3_Al_2_(SiO_4_)_3_	Ca_3_Al_2_(SiO_4_)_3−x_(OH)_4x_, 0.2 < x < 1.5	Ca_3_Al_2_(SiO_4_)_3−x_(OH)_4x_, x > 1.5	Ca_3_Al_2_(OH)_12_
Crystal System	Cubic	Cubic	Cubic	Cubic
Density	3.65	3.2	2.765	2.529
Volume	1663.7	1745.3	1887.3	1987.4

**Table 6 materials-15-08803-t006:** Corresponding wavenumbers of typical bonds.

Name	Wavenumber of Absorbance Peak (cm^−1^)
Adsorbed Water	1640 & 3450
Ca-OH in Ca(OH)_2_	3540
Ca-OH in C_2_SH	3540
γAl-OH in Hydrogrossular	3695 & 620
Si-OH in Silica Tetrahedron	1260
Si-O in C-S-H	980
Si-O-Si in C-S-H	470

**Table 7 materials-15-08803-t007:** The fitting results of the FWHM of all samples.

RH	Wavenumber	Bond	Height	Ratio b/a
5%	1049.158	a	0.01033	1.3940
	622.137	b	0.0144	
23%	1038.401	a	0.00865	1.1838
	626.247	b	0.01024	
43%	1058.536	a	0.01074	1.0214
	620.46	b	0.01097	
54%	1045.311	a	0.01215	0.8658
	631.017	b	0.01052	
75%	1064.477	a	0.01262	0.9636
	627.573	b	0.01216	
81%	1054.871	a	0.00927	0.8952
	633.608	b	0.008299	
85%	1060.836	a	0.01117	0.9017
	625.787	b	0.01008	
98%	1053.907	a	0.01423	0.9126
	618.109	b	0.01298	

a: Si–OH in [SiO_4_]^4−^; b: γAl–OH.

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
