# Peer review of "Effects of Ambient Humidity on Water Migration and Hydrate Change in Early-Age Hardened Cement Paste"

_materials, 2022, doi:10.3390/ma15248803_

Round 1
Reviewer 1 Report
The work, or rather some paragraphs, should be thoroughly changed. There are many understatements, there are no descriptions, many things are unclear and incomprehensible. The work needs to be arranged.
Additional research would also be required.

Author Response
Dear reviewer:
Please see the attachment.
Thank you for your time and valuable comments.
Best regards.

Reviewer 2 Report
See attached file.

Author Response

(The authors gave the same response as above.)

Reviewer 3 Report
This manuscript evaluates the “Effects of Ambient Humidity on Water Migration and Hydrate Change in Hardened Cement Paste”. The manuscript is elaborately described and contextualized with the help of previous and present theoretical background. All the references cited are relevant to this area of research. The methods/analytical study are clearly stated. The result and discussion section are clearly presented. The manuscript needs Minor revision and require the following modifications before the acceptance.
1. What is the research need? Mention it in the abstract. Mention the your research recommendation in the last line of abstract.
2. Relative humidity should not be abbreviated.
3. Cite the sentence ‘Cement concrete is one of the most commonly used building materials, and because of its durability and excellent compressive properties, cement concrete has been used in many extreme environments’ by using the following works.
https://doi.org/10.1016/B978-0-12-821730-6.00031-0
https://doi.org/10.3390/ma15124272
https://doi.org/10.1002/suco.201900390
https://doi.org/10.12989/sem.2022.83.3.387
4. What is the novelty of your research?
5. Include the experimental photos of the various test done.
6. Why author is not conducted tests at the age of 28 days?
7. Compare your results with existing studies.
8. What is FWHM?
9. Present your research recommendations at the end of conclusion part.
Author Response

(The authors gave the same response as above.)

Round 2
Reviewer 1 Report
Thank you for making corrections and additions. The article is now clearer.
Good luck!
Author Response
Dear reviewer:
We really appreciate your comments and help to improve our manuscript!
Thanks again!
Best regards.
Reviewer 2 Report
See attached file.

Author Response

(The authors gave the same response as above.)
